# Age of Antibiotic Resistance in MDR/XDR Clinical Pathogen of *Pseudomonas aeruginosa*

**DOI:** 10.3390/ph16091230

**Published:** 2023-08-30

**Authors:** Ashish Kothari, Radhika Kherdekar, Vishal Mago, Madhur Uniyal, Garima Mamgain, Roop Bhushan Kalia, Sandeep Kumar, Neeraj Jain, Atul Pandey, Balram Ji Omar

**Affiliations:** 1Department of Microbiology, All India Institute of Medical Sciences, Rishikesh 249203, India; ashish.sci@aiimsrishikesh.edu.in; 2Department of Dentistry, All India Institute of Medical Sciences, Rishikesh 249203, India; radhika96k@gmail.com; 3Department of Burn and Plastic Surgery, All India Institute of Medical Sciences, Rishikesh 249203, India; drvishalm@yahoo.com; 4Department of Trauma Surgery, All India Institute of Medical Sciences, Rishikesh 249203, India; drmadhuruniyal@gmail.com; 5Department of Biochemistry, All India Institute of Medical Sciences, Rishikesh 249203, India; mamgain.garima6@gmail.com; 6Department of Orthopaedics, All India Institute of Medical Sciences, Rishikesh 249203, India; roop.orth@aiimsrishikesh.edu.in; 7Department of Cellular Biology and Anatomy, Augusta University, Augusta, GA 30912, USA; sschaudhary55@gmail.com; 8Department of Medical Oncology, All India Institute of Medical Sciences, Rishikesh 249203, India; 9Division of Cancer Biology, Central Drug Research Institute, Lucknow 226031, India; 10Department of Entomology, University of Kentucky, Lexington, KY 40503, USA

**Keywords:** drug-resistant, public health concern, antibiotic stewardship, developing new antimicrobials, efflux pump systems

## Abstract

Antibiotic resistance in *Pseudomonas aeruginosa* remains one of the most challenging phenomena of everyday medical science. The universal spread of high-risk clones of multidrug-resistant/extensively drug-resistant (MDR/XDR) clinical *P. aeruginosa* has become a public health threat. The *P. aeruginosa* bacteria exhibits remarkable genome plasticity that utilizes highly acquired and intrinsic resistance mechanisms to counter most antibiotic challenges. In addition, the adaptive antibiotic resistance of *P. aeruginosa*, including biofilm-mediated resistance and the formation of multidrug-tolerant persisted cells, are accountable for recalcitrance and relapse of infections. We highlighted the AMR mechanism considering the most common pathogen *P. aeruginosa*, its clinical impact, epidemiology, and save our souls (SOS)-mediated resistance. We further discussed the current therapeutic options against MDR/XDR *P. aeruginosa* infections, and described those treatment options in clinical practice. Finally, other therapeutic strategies, such as bacteriophage-based therapy and antimicrobial peptides, were described with clinical relevance.

## 1. Introduction

There are several reasons why multidrug-resistant (MDR) and extensively drug-resistant (XDR) *P. aeruginosa* strains represent a worldwide health threat. First, *P. aeruginosa* is extremely opportunistic and can cause serious infections in hospital settings, especially in immunocompromised people. Second, it has versatility in its antibiotic resistance and may transmit drug resistance through multiple routes [1]. High-risk clones are spreading worldwide, posing a public health issue that must be examined and addressed [2]. In 2017, the World Health Organization placed carbapenem-resistant *P. aeruginosa* in the “critical” group, for which new medicines are needed [3].

*P. aeruginosa* MDR and XDR strains have increased in frequency during the past several years, with rates varying between 15 and 30% in certain geographic regions [4,5]. However, combination medication resistance is also prevalent in *P. aeruginosa* MDR and XDR strains. A total of 5.5% of *P. aeruginosa* isolates that were resistant to all five antimicrobial groups under monitoring in 2015, as well as 13% of isolates that were resistant to more than three antimicrobial groups, were reported by the European Centers for Disease Prevention and Control [6]. Recent research has examined the in vitro effectiveness of polymyxin B in combination with 13 other antibiotics (amikacin, aztreonam, cefepime, chloramphenicol, ciprofloxacin, fosfomycin, linezolid, meropenem, minocycline, rifampin, temocillin, thiamphenicol, or trimethoprim) against four clinical strains of MDR *Pseudomonas aeruginosa*. Positive interactions were frequently observed with the tested combinations against strains that carried multiple mechanisms of drug resistance, and with antibiotics that are typically inactive against *P. aeruginosa* [7]. Another study involving 3184 clinical isolates collected from 71 US medical centers shows that Ceftazidime–avibactam (97.0% susceptible (S)), ceftolozane–tazobactam (98.0% S), imipenem––relebactam (97.3% S) and tobramycin (96.4% S) were the most active agents against the aggregate *P. aeruginosa* isolate collection, and retained good activity against piperacillin–tazobactam-non-susceptible, meropenem-non-susceptible and multidrug-resistant (MDR) isolates [8]. The remedies to this issue involve investing more money in fundamental research, clinical research, antimicrobial stewardship infection control, the creation of novel antimicrobials, and the optimum use of those that are already available. In this review article, we discussed the available treatments for MDR/XDR *P. aeruginosa* infections and examined the most recent research on its resistance. We also looked at the therapeutic alternatives applied in clinical practice to treat MDR/XDR *P. aeruginosa* infections. Last but not least, several treatment modalities with clinical applicability were discovered, including bacteriophage-based therapy and antimicrobial peptide therapy, that can be used in future research and medication.

## 2. Molecular Mechanism of Drug Resistance in *P. aeruginosa*

### 2.1. Inherent Resistome

It is intriguing how *P. aeruginosa* possesses a peculiar assortment of drug-resistance mechanisms, such as multiple chromosomal-associated genes that confer resistance to antibiotics, as well as complex regulatory pathways involved in both inherent and adaptive resistance [9,10,11,12]. When compared to other Gram-negative bacteria, the development of inherent resistance in *P. aeruginosa* is primarily influenced by the expression of inducible AmpC cephalosporinase, the synthesis of constitutive and inducible efflux pumps, and the low permeability of its outer membrane. The synthesis of inducible beta-lactamase is of utmost importance in the inherent resistance of *P. aeruginosa* to aminopenicillins and cephalosporins, specifically cefoxitin. These antibiotics are potent inducers of AmpC expression, resulting in an overabundance of cephalosporinase [13]. The hydrolytic stability of Imipenem is slightly affected by its strong ability to induce enzymes. The expression of inducible AmpC is important in reducing the natural sensitivity of *P. aeruginosa* [14]. Additionally, the recently discovered imipenemase (IMP) PA5542 may also influence the inherent susceptibility of β-lactam antibiotics [15]. Nevertheless, additional research is needed to explore their involvement in innate or acquired resistance. The expression of efflux pump is crucial in reducing the inherent susceptibility to a wide range of β-lactam antibiotics and fluoroquinolones. In a similar manner, the inducible expression of Hyperproduction of efflux-mediated (MexXY) genes exerts a notable impact on the inherent, minimal resistance to aminoglycosides [16]. These efflux pump systems effectively expel various classes of antibiotics from the bacterial cell, thereby providing it with intrinsic resistance.

### 2.2. Mutational Resistome

In addition to its wide-ranging innate resistance repertoire, *P. aeruginosa* demonstrates remarkable proficiency in acquiring chromosomal mutations, leading to the emergence of novel antimicrobial-resistant superbugs [1], as summarized in Table 1. The β-lactam resistance mechanism driven by mutation has been observed in 20% of *P. aeruginosa* [12,17,18]. The deactivation of penicillin binding protein-4 (PBP-4) triggers the activation of the CreBC/BlrAB two-component system, which is responsive to carbon sources and contributes to β-lactam resistance. This activation leads to an additional increase in resistance levels [18]. The clinical strains were found to possess distinct mutations that affected the transcriptional regulator of AmpR, a protein responsible for regulating the overexpression of *ampC* and conferring resistance to beta-lactam antibiotics. The mutations under consideration encompass the R154H mutation, which is associated with the epidemic MDR/XDR ST175 high-risk clone, and the D135N mutation, observed in species other than *P. aeruginosa* [12]. Numerous genetic variations have been documented to enhance the amplification of ampC in various genetic sequences, encompassing those responsible for other amidases (*AmpDh2/AmpDh3*), *PBP5* or *PBP7*, lytic transglycosylases (*MltB* and *SltB1*), *MPL* (*UDP-N-acetylmuramate*: *L-alanyl—D-glutamyl-Meso-(NADH dehydrogenase I chain N*) [13].

Findings from studies have demonstrated that genetic variations that modify the structure of *AmpC* can lead to the development of resistance against β-lactam antibiotics. This resistance includes the newly developed β-lactam–lactamase inhibitor, as well as the combinations of ceftolozane–tazobactam and ceftazidime–avibactam. This is in addition to the phenomenon of *AmpC* hyperproduction, as previously reported [19,20,21,22]. The development of resistance to ceftolozane–tazobactam and ceftazidime–avibactam was observed in a specific group of *P. aeruginosa* clinical isolates [23]. This resistance was linked to various changes in the amino acid composition of *AmpC*. Recent findings have unveiled the existence of more than 300 distinct variations of cephalosporinase derived from the *Pseudomonas* genus. Notably, certain variations have been observed to confer increased resistance to ceftolozane–tazobactam and ceftazidime–avibactam. There is a growing body of evidence suggesting that alterations in penicillin-binding proteins (PBPs), specifically mutations in PBP-3, contribute to the development of resistance to β-lactam antibiotics, alongside β-lactamases. Recent data obtained from individuals diagnosed with cystic fibrosis [24,25], as well as from strains of bacteria causing epidemics [26,27], and laboratory experiments conducted in controlled environments [28,29], have provided evidence indicating that specific alterations in penicillin-binding protein-3 play a role in the emergence of resistance to -lactam antibiotics. The *R504C/R504H* and *F533L* mutations, located in the domains responsible for stabilizing the -lactam–penicillin-binding protein-3 inactivation complex, have been frequently documented in the scientific literature [30]. The presence of inhibitory deletion/insertion sequences within the *OprD* gene, as well as distant mutations that enhance the activity of efflux pump systems *MexEF-OprN* or *CzcCBA* while simultaneously reducing the expression of *OprD*, can result in the loss of the carbapenem-specific porin—*OprD*. The inactivation of *OprD* often leads to resistance against all conventional anti-pseudomonal β-lactams in a synergistic fashion when combined with *AmpC* overexpression [31]. Another pivotal determinant in resistance is the mutational upregulation of one of the four primary efflux pumps in *P. aeruginosa* [32,33]. The prevalence of *MexAB*-*OprM* and *MexXY* overexpression in clinical isolates ranges from 10% to 30%, while the overexpression of *MexCD-OprJ* and *MexEF-OprN* is less-frequently observed, occurring in approximately 5% of cases. The *OprD* porin exhibits either inactivity or downregulation, leading to reduced sensitivity to Meropenem and the inducible synthesis of *AmpC*. The simultaneous upregulation of *MexAB-OprM* and downregulation of *OprD* is a significant determinant of Meropenem resistance in clinical strains [34].

More than 20% of isolates frequently demonstrate resistance to imipenem, and the majority of these isolates are deficient in *OprD* [34]. The *MexAB-OprM* efflux pump exhibits the most extensive substrate specificity, and its upregulation due to mutations leads to decreased susceptibility to all -lactams and fluoroquinolones (with the exception of imipenem). The mutation-driven overexpression of *MexXY* is a common contributing factor in the resistance of clinical strains to cefepime, in addition to its primary role in intrinsic aminoglycoside resistance [35]. The hyperproduction of *MexEF-OprN* is not commonly observed and primarily impacts fluoroquinolone antibiotics. However, mutations in *mexT/mexS* genes that lead to *MexEF-OprN* hyperproduction also result in resistance to imipenem by repressing *OprD* gene expression [36]. In spite of exhibiting heightened resistance to various β-lactams and aminoglycosides, the upregulation of *MexCD-OprJ*, a phenomenon frequently observed in persistent infections, additionally plays a role in conferring resistance to cefepime [37].

Fluoroquinolone resistance in *P. aeruginosa* often arises due to the overexpression of efflux pumps, as well as mutations occurring in type IV topoisomerases (*ParC* and *ParE*) and DNA-gyrases (*GyrA* and *GyrB*) [38]. From a geographical perspective, the prevalence of fluoroquinolone resistance is observed to be the dominant trait, ranging from 30% to 40% in multiple countries. Recent scientific investigations have elucidated that genetic mutations occurring in the *fusA1* gene, responsible for encoding the elongation factor G, have the potential to induce resistance to aminoglycoside antibiotics. This resistance mechanism operates in conjunction with the overexpression of the MexXY genes and the acquisition of genetic pathways through horizontal gene transfer. Indeed, empirical evidence has shown that certain mutations in the *FusA1* gene can result in resistance to aminoglycoside antibiotics in laboratory settings [39,40] and in clinical cases of *P. aeruginosa* infection, especially in individuals with cystic fibrosis [41].

In essence, although the occurrence of colistin resistance is still relatively limited (5%), there has been a recent surge, likely due to its heightened utilization as a final option against infections caused by multidrug resistant/extensively drug resistant bacterial strains. The development of colistin resistance frequently occurs due to alterations in the lipid A component of lipopolysaccharide (LPS) following the addition of 4-amino-4-deoxy-L-arabinose [42]. The mutations observed are often associated with the regulatory systems *PmrAB* and *PhoPQ*, which result in the activation of the arnBCADTEF operon. In recent studies, it has been demonstrated that mutations in the ParRS two-component regulator play a crucial role in driving colistin resistance. These mutations activate the *arnBCATEF* genes, leading to an MDR profile. Furthermore, they upregulate the *MexXY* genes while downregulating the *OprD* gene [11]. The *ColRS* and *CprRS* systems have also been implicated in the development of polymyxin resistance [43].

### 2.3. Horizontally Acquired Resistome

Novel antibiotics of the latest generation have been developed with the specific purpose of selectively inhibiting crucial cellular proteins involved in DNA replication and repair, protein synthesis, and the production of components for the cell membrane [44]. The primary strategies employed to address acquired resistance involve implementing chemical modifications to preexisting antibiotics. The rate of antibiotic production has experienced a notable decrease in recent years, despite the fact that a considerable number of antibiotics are presently undergoing their third or fourth round of modifications. Furthermore, due to the evolutionary adaptability of bacteria, the efficacy of antibiotic therapy has progressively diminished over a period of time [45]. Furthermore, bacteria have developed a complex regulatory evolutionary adaptation by acquiring resistance genes primarily through conjugation and, to a lesser degree, through spontaneous transformation and transduction [46]. Despite the perceived insignificance of transformation, recent research suggests that its importance may be greater than previously hypothesized [47]. Recent research examined the effectiveness of horizontal gene transfer (HGT) through conjugation and examined the MDR phenotypes of numerous clinical and environmental bacterial strains from various sources. Along with examining the effects of medications and heavy metal (arsenic), conjugation efficiency between clinical and environmental strains was also examined. They discovered that using 2-HDA as a COIN prevented HGT between strains that were obtained in hospitals and those that naturally exist [48].

One aspect of mutational resistance that is of significant interest is the transferable type of *P. aeruginosa* resistance, which occurs relatively frequently and contributes to the overall accumulation of concern. Bacterial conjugation serves as the fundamental mechanism for both intra- and inter-species HGTs. It plays a crucial role in expediting the dissemination of antibiotic resistance genes [49]. Certainly, the prevalence of highly hazardous transferable β-lactamases, including *ESBLs* and carbapenemases (specifically *class B carbapenemases*, also known as *Metallo β-lactamases*), is steadily increasing on a global scale. Nevertheless, their distribution exhibits inconsistency and displays variation across hospitals and regions, ranging from less than 1% to approximately 50% [50]. Moreover, the occurrence of transferable *β-lactamases* in *P. aeruginosa* might have been underestimated in several locations due to the challenges associated with their detection [51]. Integrons belonging to Class 1 generally encompass determinants of resistance to aminoglycosides, as well as the genes responsible for extended-spectrum beta-lactamases (ESBLs) and carbapenemases. While the involvement of conjugative elements is now more commonly recognized, these integrons are often inserted into transposable elements located on the bacterial chromosome [51,52,53,54]. A recent study was conducted to review the distribution of spreadable β-lactamases in *P. aeruginosa* [55]. Frequently documented extended-spectrum beta-lactamases (ESBLs) in *P. aeruginosa* encompass class D enzymes, specifically OXA-2 or OXA-10 variants, as well as class A enzymes including PER, VEB, GES, BEL, and PME variants. Metallo β-lactamases (MBLs) are the predominant carbapenemases found in *P. aeruginosa*. Among these MBLs, the VIM and IMP variants are the most prevalent and widely distributed across different geographical regions. In Brazil, the prevalence of the *SPM MBL* gene is extensive, while the *NDM*, *GIM*, and *FIM* genes are sporadically detected. The prevalence of Class A carbapenemases in *P. aeruginosa* is relatively low on a global scale, even though GES and KPC enzymes have been identified in multiple countries [54].

The transferability resistance of aminoglycosides is influenced by the presence of aminoglycoside-modifying enzymes that are encoded within Class 1 integrons. The acetyltransferases frequently observed in *P. aeruginosa* are AAC 3′ gentamicin and AAC 6′ tobramycin, as well as the nucleotidyltransferase ANT 2′-I gentamicin and tobramycin. Nevertheless, there are significant emerging concerns associated with 16S rRNA methyltransferases, such as *Rmt* or *Arm*, as they confer resistance to all commercially available aminoglycosides, including the recently developed plazomicin [54]. Intermittently, it has been observed that the prevalence of transferable resistance to fluoroquinolones is primarily influenced by *Qnr* determinants, such as *QnrVC1* [56]. In a recent scientific study, it has been demonstrated that a novel phosphotransferase, known as *CrpP*, is responsible for facilitating plasmid-mediated quinolone resistance [57].

Ceftolozane–tazobactam and Ceftazidime–avibactam, two recently developed combinations, exhibit a notable degree of resistance to AmpC hydrolysis [58,59]. This resistance is attributed to the inhibitory effect of avibactam on AmpC in the case of ceftazidime–avibactam, and the ability of ceftolozane to remain stable against hydrolysis by AmpC in the case of ceftolozane–tazobactam. Nevertheless, based on existing in vitro and in vivo research, it appears that the emergence of resistance to both drugs could be attributed to a combination of genetic mutations, leading to increased production of AmpC and alterations in its structure [19,20,22]. The empirical evidence obtained from experiments conducted in living organisms (in vivo) and in controlled laboratory conditions (in vitro) suggests that particular genetic alterations in penicillin-binding protein-3 have the potential to reduce the vulnerability to the aforementioned combinations. The susceptibility of ceftazidime–avibactam seems to be more influenced by the overexpression of different efflux pumps compared to ceftolozane–tazobactam [23,60]. Both ceftolozane–tazobactam and ceftazidime–avibactam have demonstrated a lack of efficacy against strains that produce acquired β-lactamases. Ceftazidime–avibactam, but not ceftolozane–tazobactam, exhibits potential activity against isolates that generate class A carbapenemases, such as GES enzymes [61]. In a similar vein, the effectiveness of ceftolozane–tazobactam and ceftazidime–avibactam against strains of *P. aeruginosa* that produce extended-spectrum beta-lactamase (ESBL) exhibits variability, albeit with a generally favorable outcome for ceftazidime–avibactam. Ultimately, the development of resistance to both pharmaceutical agents can arise due to the presence of extended-spectrum mutations in horizontally transferred *OXA-type β-lactamases* [62,63].

### 2.4. Antibiotic Resistance by SOS Response

A universally preserved bacterial stress response is primarily triggered by DNA damage. The SOS response initiates and coordinates various biological processes, including DNA repair mechanisms, bacterial cell division arrest, and latent bacteriophage induction. The SOS response is characterized by the activation of DNA polymerases IV and V. This occurs when the DNA damage is prolonged and significant. (Figure 1) depicts the visual representation of the data or information being discussed. Bacterial cultures cultivated in an artificial environment were employed in nearly all studies pertaining to the SOS response and the development of resistance to antibiotics. Prior studies have established a correlation between the mutator phenotype and the ability of bacteria, such as *P. aeruginosa*, to cause chronic infections in individuals with cystic fibrosis [64,65]. Research examining the genetic alterations in consecutively recovered strains has provided evidence supporting the activation of the SOS response in vivo [66]. Additionally, other indirect evidence has also been reported [67,68].

#### SOS-Dependent Mutagenesis and Resistance

The cellular SOS response is a captivating bacterial defense mechanism through which bacteria can acquire drug resistance, induce mutagenesis, and undergo genome reorganization [69]. The Lexi promoter-binding repressor protein regulates the SOS system. Upon binding to the RecA filament, the LexA protein undergoes self-cleavage, resulting in a reduction in LexA protein levels within the cell. This process subsequently triggers the activation of over 40 genes in bacterial cells, including the *recA* gene [70]. The proteins associated with the SOS response play a crucial role in regulating various metabolic processes within bacterial cells [71]. Furthermore, mutagenesis is initiated during the advanced phases of the SOS response. The PolV polymerase has been identified as a key driver of SOS-dependent mutagenesis in *Escherichia coli* (*E. coli*) [72]. The polymerases known for their high error rates in DNA synthesis, which are notorious for their low accuracy, encompass PolV polymerase within their category. One of the causes of induced mutagenesis can be attributed to the activity of PolV, which leads to the insertion of an erroneous nucleotide into the DNA molecule. The formation of a RecA by PolV polymerase is expected to impose certain constraints on the potential diversity of recombinases during the process of selection [73]. The process of replicating damaged DNA, known as translesion synthesis (TLS), entails the utilization of PolV polymerase to bypass DNA lesions.

While pol II, pol IV, and polV polymerase are involved in TLS, they also exert an inhibitory effect on RecA-dependent recombination. Achieving equilibrium between these two strategies is of utmost importance. The TLS phenomenon accounts for approximately 1% to 2% of the occurrences in the absence of the SOS response. According to the TLS mechanism, it has been observed that when subjected to stress, TLS has the potential to increase by up to 40% [74]. If recombination is performed by specific *RecA* variants that are efficient in polymerizing onto single-stranded DNA but somewhat impaired in strand invasion, the ratio may also significantly shift in favor of the TLS mechanism [75]. Simultaneously, bacteria experience significant detrimental effects due to the rise in mutations. As recombination decreases, there is a subsequent decrease in the size of the bacterial population [75]. Moreover, there exists a possibility that moderately unfavorable mutations may undergo fixation when the magnitude of the bacterial population is significantly diminished due to stochastic genetic drift. DNA recombination plays a crucial role in impeding detrimental mutations, as it establishes the boundaries that prevent the occurrence of a “mutational catastrophe” [76].

There exist alternative mechanisms for induced mutagenesis, notwithstanding the fact that PolV polymerase (*UmuD2C*) is the conventional origin of mutations for the purpose of evolutionary selection. While the RecA protein does not engage in interactions to generate a mutasome, an additional error-prone E. coli Pol IV polymerase is synthesized during the SOS response [77]. Both polymerases are widely distributed among the majority of bacterial species and belong to the Y family [78]. In spite of the substantial diversity observed within the Y family, it is noteworthy that the majority of polymerases share a conserved sequence of 30 residues at their C-terminus. Induced mutagenesis has been observed to occur through the activity of closely related families of polymerases in various bacterial taxa. *DnaE2* polymerase, classified as a member of the C family of polymerases, plays a crucial role in the emergence of evolutionary resistance in the bacterium *Mycobacterium tuberculosis* [79].

### 2.5. Biofilm-Mediated Resistome

The sensitivity of pseudomonas cells cultivated in biofilms is comparatively lower to antimicrobial agents and host immune responses when compared to cells grown in free aqueous suspension [80]. When bacteria proliferate within a biofilm, even those lacking protective mutations or innate resistance mechanisms can exhibit a reduced susceptibility to antibiotics [81]. When bacteria experience a loss of biofilm protection, there is a rapid restoration of antibiotic sensitivity. This suggests that the resistance to antibiotics mediated by biofilms is not a result of genetic changes or an adaptive mechanism [82]. The overarching mechanisms underlying biofilm-mediated resistance involve impeding the penetration of antibiotics, creating a modified microenvironment that hinders the growth of biofilm cells, triggering an adaptive stress response, and promoting the differentiation of persister cells. These processes collectively serve to safeguard bacteria from the detrimental effects of antibiotic treatment [81].

*P. aeruginosa* synthesizes DNA, proteins, and exopolysaccharides, which are utilized for the formation of a biofilm on the surfaces of lung epithelial cells, leading to the development of persistent lung infections [83]. The development of *P. aeruginosa* biofilms is regulated by multiple factors, primarily including quorum-sensing systems, the *GacS/GacA* and *RetS/LadS* two-component regulatory systems, exopolysaccharides, and c-di-GMP [84]. Bacterial communication, also known as quorum sensing, regulates the expression of genes in response to changes in the number of cells present [85]. *P. aeruginosa* exhibits three prominent quorum-sensing systems, namely LasI-LasR, RhlI-RhlR, and PQS-MvfR, which collectively contribute to the formation of fully developed and specialized biofilms [86]. The GacS/GacA system was found to play a beneficial regulatory role in biofilm development, as evidenced by a tenfold reduction in biofilm generation in a GacA-deficient strain of *P. aeruginosa* (PA14) compared to the wild-type PA14 strain (Figure 2) [87]. The acidification of the environment and upregulation of genes controlled by the PhoPQ and PmrAB two-component regulatory systems were observed as a result of the presence of *P. aeruginosa’s* environmental DNA (eDNA). This led to a notable increase in aminoglycoside resistance, indicating a previously unknown function of eDNA [88]. The intracellular molecule known as c-di-GMP serves as a nucleotide second messenger in the process of signal transduction [89]. It plays a role in increasing the levels of c-di-GMP within cells, and these levels are associated with the development of biofilms. In contrast, a diminished level of c-di-GMP has been found to be associated with the presence of planktonic cells [90].

Throughout the process of biofilm formation, the bacterium *P. aeruginosa* undergoes numerous changes in its physiological and phenotypic characteristics [91]. For example, strains of *P. aeruginosa* undergo a transformation into a mucoid phenotype during chronic infection in individuals with cystic fibrosis (CF). This transformation is characterized by an enhanced production of alginate, which is stimulated by the specific conditions present in the CF environment. This increased alginate synthesis facilitates the formation of biofilm colonies by the bacteria [92]. The periplasmic cyclic β-(1,3)-glucans, with which tobramycin had physical interaction and were sequestered in the periplasm prior to reaching its target site, were produced through the activity of the glucosyltransferase encoded by the *ndvB* gene [93]. An operon encompassing the gene *PA14 40260-40230* encodes a novel efflux pump. The resistance of *P. aeruginosa* to gentamicin and ciprofloxacin in biofilm was observed to decrease upon deletion of the specific operon [94]. The regulation of Type VI secretion in *P. aeruginosa* is governed by the *tssC1* gene, which exhibits a high level of expression within biofilm structures [95].

## 3. MDR/XDR *P. aeruginosa* Epidemiology

The MDR phenomenon is characterized by the lack of susceptibility to at least one agent in a minimum of three antibiotic classes. XDR is defined as the lack of susceptibility to at least one agent in all but one or two antibiotic classes. Pan-drug resistance (PDR) is the state of being unable to be susceptible to all agents in all antibiotic classes. The subsequent suggested antibiotics for testing antipseudomonal antibiotics are cephalosporins (ceftazidime and cefepime), antipseudomonal penicillin’s combined with β-lactamase inhibitors, monobactams, antipseudomonal carbapenems, aminoglycosides, fluoroquinolones, phosphonic acids, and polymyxins. Although the aforementioned suggestion undeniably holds value in terms of standardizing the descriptions of *P. aeruginosa* resistance profiles, it is imperative to consider various additional factors. The outcome will exhibit variability based on the utilization of either EUCAST or CLSI breakpoints, even when employing a singular definition. The extensive implementation of the suggested definition is constrained by the absence of clinical breakpoints (as defined by CLSI and EUCAST) for one of the substances (fosfomycin). In a similar vein, it was previously believed that EUCAST breakpoints for aztreonam were not applicable to criteria for MDR, XDR, and PDR bacteria due to acquired resistance. This was due to the inherent resistance of *P. aeruginosa* to aztreonam. The current classification does not consider newly developed antipseudomonal medications such as ceftazidime–avibactam and ceftolozane–tazobactam [56,96]. Moreover, a significant number of MDR strains meet the criteria for extensively XDR strains, thereby further constraining the spectrum of viable therapeutic options. In a study conducted in 2017, it was found that 26% of *P. aeruginosa* infections in Spain exhibited MDR. Furthermore, 65% of these MDR isolates (equivalent to 17% of all isolates) met the criteria for XDR classification. The investigation was carried out on a significant scale, encompassing 51 hospitals, utilizing a multicenter approach. The vast majority of isolates exhibited susceptibility to either amikacin or colistin, with a total of 102 isolates falling into this category. Indeed, the prevalence of colistin-only sensitive (COS) profiles is high in numerous hospitals worldwide, and the existence of pan-drug resistance has already been documented [97,98]. Nevertheless, the presence of resistance to novel antipseudomonal drugs did not play a significant role in the majority of research studies. The geographical distribution of acquired beta-lactamases, including ESBLs or carbapenemases, exhibits significant variation, despite the overall low prevalence of resistance to these innovative therapeutic options, typically below 10% [98,99,100,101].

## 4. Clinical Impact of MDR *P. aeruginosa*

Selecting the optimal empirical antibiotic for the management of MDR pathogens can pose a challenging task. Individuals afflicted with MDR/XDR infections exhibit a heightened propensity towards receiving inadequate initial antimicrobial therapy [102,103]. In individuals afflicted with bloodstream infections caused by *P. aeruginosa*, the postponement of commencing appropriate antibiotic treatment has been found to be associated with unfavorable outcomes and increased mortality rates [104,105,106]. In the context of these infections, the presence of MDR/XDR patterns is associated with an increased likelihood of receiving inadequate empirical treatment based on evidence [107,108,109,110]. In a similar vein, antimicrobial medications that are administered as second or third-line treatments are commonly utilized as directed therapy for infections caused by MDR/XDR pathogens. As a result, their performance is inferior to that of medications utilized in the treatment of infections caused by susceptible strains [111]. However, the causal relationship between multidrug resistance and clinical outcomes remains unclear. The poorer prognosis can be attributed to the fact that the colonization and infection of MDR/XDR *P. aeruginosa* usually manifests in patients with multiple underlying medical conditions [112,113]. The mortality of individuals may be attributed to elevated levels of pre-existing comorbidities. There is a growing body of research focused on investigating the biological impacts of antibiotic resistance on the pathogenicity of *P. aeruginosa*. In the scientific community, there is a prevailing belief that the emergence of resistance mechanisms is associated with fitness expenses that diminish the pathogenicity of MDR/XDR strains [114]. Nevertheless, specific resistance mutations have also been observed to exhibit no discernible correlation with reductions in fitness [115,116]. Based on scientific research, it has been observed that MDR strains have the ability to develop suppressor or compensatory changes. These changes allow them to restore their original level of fitness, thereby preventing them from experiencing a decline in their virulence over time [115,117]. As previously stated, *P. aeruginosa* possesses numerous virulence factors [118,119]. The secretion system plays a crucial role in determining the pathogenicity of microorganisms [120,121]. For instance, the type III secretion system (T3SS), also known as the T3SS, is responsible for injecting effector cytotoxins (ExoS, ExoT, ExoU, and ExoT) into host cells [119,121]. The upregulation of ExoU, the most potent among the four identified effector exotoxins, has been correlated with an unfavorable prognosis [122]. The presence of the exoU+ genotype was found to be associated with increased early mortality in a recent clinical trial involving individuals diagnosed with *P. aeruginosa* bacteremia. It was postulated that this genotype could potentially serve as a predictive biomarker for *P. aeruginosa* infections [4]. Additional virulence factors of *P. aeruginosa* have been recently discovered, including the toxin ExlA. This toxin induces rupture of the plasma membrane of host cells, thereby enhancing the pathogenicity of the bacterium [123]. In their study, Pea et al. [4] identified a correlation between certain type III secretion system (TTSS) genotypes and patterns of antibiotic resistance. Specifically, they found that the exoU+ genotype was less prevalent in MDR strains of *P. aeruginosa*. This finding sheds light on the impact of multidrug resistance on the pathogenicity of *P. aeruginosa* [4]. Furthermore, multiple studies indicate a correlation between specific resistance profiles and the Type Three Secretion System (TTSS) [122]. The exoU+ genotype is exclusively found in one out of the three most high-risk clones globally, namely ST235 [4]. Nevertheless, numerous experimental and clinical studies indicate a potential decrease in virulence in multidrug resistant/*P. aeruginosa* [4,110,124]. In vitro investigations have demonstrated that MDR strains exhibit a slower development rate and display deficiencies in virulence determinants, such as bacterial motility or pigment synthesis [125]. Resent study show the role of pigment production in antibiotic resistance, Yellow pigment-producing *P. aeruginosa* strains posed a significant problem, which may be associated with the development of multidrug resistance [126]. *P. aeruginosa* MDR/XDR strains have demonstrated reduced capacity compared to susceptible bacteria in terms of causing infection, eliciting an inflammatory response, and resulting in mortality in animal models conducted in vivo [125,127]. The reduced pathogenicity of MDR *P. aeruginosa* strains is supported by clinical research [4,110,124]. As previously mentioned, it has been observed that at least one of the international XDR high-risk clone strains exhibits sustained high virulence irrespective of its resistance profile. Consequently, it is advisable to exercise caution when interpreting the data. In order to enhance clarity, further investigation is necessary.

## 5. Antibiotic Agents for of MDR *P. aeruginosa*

### 5.1. Polymyxins

Colistin (polymyxin E) and polymyxin B are the two polymyxins that are commonly employed in clinical environments. However, a significant amount of preclinical and clinical data pertaining to these “ancient medications” has recently been revealed [128,129,130,131]. The antibacterial efficacy of polymyxins is contingent upon their chemical composition. Moreover, the hydrophobic regions of polymyxins have the ability to interact with the lipopolysaccharide (LPS) [132]. The interactions mentioned [132,133] lead to the disruption of the bacterial cell membrane.

Nevertheless, the exact mechanism behind bacterial cell death remains to be fully comprehended [134]. Recent scientific investigations on *P. aeruginosa* have presented findings that question the established belief regarding the mechanism of action of colistin, which was previously thought to eliminate bacteria by causing damage to the cytoplasmic membrane [135,136,137]. Additional hypotheses, such as the proposition that bacterial demise occurs through phospholipid interchange between the outer and cytoplasmic membranes, impeding respiratory enzymes and generating reactive oxygen species, warrant further exploration in subsequent scientific investigations [137].

Colistin is administered in the form of an inactive prodrug known as colistin methane sulfonate (CMS), while Polymyxin B is directly administered as an active antibiotic. However, it should be noted that after injection, Polymyxin B needs to undergo a conversion process to become colistin [138]. The frequency of usage of a specific polymyxin may vary depending on the geographical region. After administering a loading dosage of 9 million international units (IU), several clinical investigations were conducted to evaluate the efficacy of parenteral colistin at higher doses (4.5 IU administered every 12 h) [139,140]. The clinical outcomes of patients who received doses calculated using the Garonzik et al. equation have not been supported by any clinical data [129]. In 2016, Nation et al. [131] revised this equation, and Sorl et al. conducted a study to examine the impact of colistin plasma concentrations on clinical outcomes in 91 patients with infections caused by MDR *P. aeruginosa*. The aim of their study was to apply pharmacokinetics and pharmacodynamics (PK/PD) information to clinical practice [141].

In summary, the absence of clinical trials assessing the outcomes of individuals adhering to updated guidelines [131,142] leaves a gap in evaluating the efficacy of higher colistin doses, as suggested by PK/PD studies. Colistin serves as a viable alternative for the treatment of urinary tract infections due to the presence of substantial amounts of recently synthesized colistin in urine [142]. Nevertheless, there are a lack of clinical data to substantiate this decision. Colistin has been the focus of numerous published clinical investigations for the management of MDR/XDR *P. aeruginosa* infections. The majority of studies consist of single-center retrospective series with small patient populations, with the exception of two outliers that each involve more than 100 patients [143,144]. In the context of MDR/XDR *P. aeruginosa* infections, the utilization of colistin in combination with antibiotic therapy has been demonstrated to be a more favorable treatment modality [145]. Another crucial inquiry regarding the utilization of polymyxins in the management of multidrug-resistant/*P. aeruginosa* infections is whether the concurrent administration of multiple therapies can improve patient outcomes.

The clinical use of polymyxin has just been resumed. SPR741 is a polymyxin B derivative with less nephrotoxicity. SPR741 does not directly kill bacteria, but it improves the efficiency of antibiotics that are co-administered because they would not otherwise reach their intracellular targets [146]. A total of 64 healthy adult volunteers were included in a phase I randomized control experiment to assess the safety, tolerability, and pharmacokinetics of both single and multiple intravenous doses of SPR741 (NCT03022175) [147].

### 5.2. Carbapenems

Various in vitro and in vivo investigations have established that pharmacokinetics/pharmacodynamics (PK/PD) is the most reliable method for determining the proportion of the dosing interval during which the concentration of unbound or free drug in the bloodstream surpasses the minimum inhibitory concentration (MIC) required to combat the pathogen. A pharmacodynamics study employed Monte Carlo simulation techniques to assess various dosage regimens of meropenem, administered through intermittent or extended infusions, in their effectiveness against Enterobacteriaceae and *P. aeruginosa* strains with varying susceptibilities. A total of 276 isolates of *P. aeruginosa* were included in the study, with 22.1% of them exhibiting minimum inhibitory concentration (MIC) values greater than 4 mg/L. To achieve a 50% probability of maintaining a free drug concentration above the minimum inhibitory concentration (MIC) required to effectively combat all susceptible *P. aeruginosa* isolates (with MIC values of 4 mg/L or lower), a dosage of 1 g of meropenem every 8 h in extended infusion or 2 g every 8 h in intermittent/extended infusion is necessary. Nevertheless, in the case of organisms that are classified as intermediate-resistant to meropenem (with a minimum inhibitory concentration of 8 mg/L), it was observed that only the higher-dose regimen of 2 g administered every 8 h through extended infusion was able to provide sufficient bactericidal exposure. The researchers proposed that the administration of meropenem at the highest dose of 2 g/8 h through extended infusion could be an effective treatment for *P. aeruginosa* strains that are classified as intermediate or resistant [148]. A study was conducted using an in vitro infection model and Monte Carlo simulation to assess the most effective dosage of imipenem in combination with tobramycin against clinical isolates of *P. aeruginosa* that are resistant to carbapenems and aminoglycosides. The optimal antibacterial activity of imipenem was observed at simulated doses of 4 or 5 g/day administered via continuous infusion, in combination with tobramycin [149]. The adequacy of this dosage regimen in a neutropenic mouse thigh model of XDR *P. aeruginosa* infection was confirmed by the same group [149]. In a single study, a total of 237 cases of bloodstream infections caused by *P. aeruginosa* with reduced susceptibility to carbapenem were analyzed. The study aimed to evaluate the effectiveness of different dosing regimens of carbapenem antibiotics, specifically imipenem, meropenem, and doripenem. The dosing regimens tested included imipenem administered at a dose range of 0.5 to 1 g every 6 h using both 0.5- and 3 h infusion durations, meropenem administered at a dose range of 1 to 2 g every 8 h using both 0.5- and 3 h infusion durations, and doripenem administered at a dose range of 0.5 to 2 g every 8 h using both 1- and 4 h infusion durations. A T > MIC value of 40% was determined to be the optimal pharmacokinetic/pharmacodynamic ratio. The findings indicated that the administration of meropenem at a dosage of 2 g every 8 h over a 3 h infusion period, and doripenem at a dosage of 1 g every 8 h over a 4 h infusion period, exhibited the highest effectiveness in combating *P. aeruginosa* strains that displayed decreased susceptibility to carbapenem antibiotics [150]. In a case report involving a critically ill patient who underwent a double-lung transplant and developed pneumonia caused by multidrug resistant *P. aeruginosa* with a meropenem minimum inhibitory concentration (MIC) of 32 mg/L, the patient was treated with a continuous infusion of meropenem at a dosage of 8 g per 24 h. Remarkably, this treatment approach resulted in the successful resolution of the patient’s clinical condition [151]. The occurrence of carbapenem-resistant *P. aeruginosa* (CRPA) primarily arises due to the presence of carbapenem-hydrolyzing enzymes. Metallo-beta-lactamases (MβLs), such as Verona integron-encoded metallo-beta-lactamases (VIMs), IMPs, and New Delhi metallo-beta-lactamases (NDMs), are widely recognized as crucial enzymes responsible for antibiotic resistance in clinical pathogens of *P. aeruginosa*.

### 5.3. Antipseudomonal β-Lactams

Limited data are currently accessible regarding the efficacy of various conventional antipseudomonal β-lactams, including cefepime, ceftazidime, piperacillin–tazobactam, and aztreonam, as standalone treatments for MDR/XDR *P. aeruginosa* infections. Aztreonam therapy could potentially serve as a viable treatment for Ambler class B metallo-beta-lactamase (MBL)-producing Gram-negative bacteria (GNB), including *P. aeruginosa*. A study was conducted to assess the clinical efficacy of a particular series in treating infections caused by *P. aeruginosa* that produce metallo-beta-lactamase (MBL). The observed mortality rate in the conducted trial was determined to be 30%. The sample size utilized in the study was insufficient to establish definitive conclusions, with the majority of cases [152] involving the administration of combination therapy. Out of the nine patients infected with MBL-producing Pseudomonas who were treated with intravenous colistin in combination with either aztreonam or piperacillin–tazobactam, a total of seven patients (77.8%) experienced favorable outcomes and survived [153]. According to a clinical case study [154], a patient with a compromised immune system and a wound infection caused by multidrug resistant *P. aeruginosa* was effectively treated using a continuous infusion of high-dose aztreonam at a rate of 8.4 g per day. A critically immunocompromised patient with multidrug resistant *P. aeruginosa* bacteremia and a ceftazidime minimum inhibitory concentration (MIC) of 64 mg/L [155] was effectively administered a continuous infusion of high-dose ceftazidime ranging from 6.5 to 9.6 g per day. As previously mentioned, certain in vitro combination assays, specifically those involving cefepime–tobramycin [156] and cefepime–aztreonam [157], have demonstrated additive or synergistic effects against *P. aeruginosa*. Based on the specific strain, resistance phenotype, and genotype, a potential approach in individual cases may involve the utilization of high doses of certain drugs delivered through prolonged infusion as part of a combination therapeutic regimen.

Relebactam, a newly developed beta-lactamase inhibitor, specifically targets various beta-lactamase enzymes in multidrug-resistant bacteria [158]. Imipenem with relebactam against *P. aeruginosa* was introduced. As a result, relebactam’s effectiveness against *P. aeruginosa* has been the subject of numerous clinical trials (NCT02493764, NCT02452047, NCT05561764, NCT03583333, and NCT05204563. In a subsequent phase III clinical trial (NCT02493764 NCT03583333), patients with ventilator-associated or hospital-acquired bacterial pneumonia were randomized to receive either piperacillin/tazobactam intravenously every six hours for seven to fourteen days, or imipenem/cilastatin/relebactam. Treatment included either the fixed-dose combination of imipenem/relebactam/cilastatin, or the fixed-dose combination of piperacillin/tazobactam. The 264 patients who received imipenem/cilastatin/relebactam and the 267 patients who received piperacillin/tazobactam both showed similar improvements in mortality, morbidity, and clinical symptoms. Imipenem/cilastatin/relebactam can effectively treat patients with Gram-negative bacterial pathogen infections, including *P. aeruginosa*, even in critically ill, high-risk patients, according to the study’s overall findings [159]. Additionally, nacubactam, a different new beta-lactamase antibiotic, has demonstrated encouraging outcomes as a strong antibiotic against *P. aeruginosa* infections (NCT02134834, NCT02972255).

### 5.4. Aminoglycosides

Certain aminoglycosides [158,159] are administered in conjunction with other antimicrobial agents to address highly resistant infections caused by multidrug-resistant/*P. aeruginosa*. Multiple pharmacodynamics, in vitro and in vivo investigations have confirmed that aminoglycosides exhibit antibacterial activity that is dependent on the concentration, and a peak concentration of ≥8 to 10 is the most reliable predictor of efficacy in terms of pharmacokinetics/pharmacodynamics (PK/PD) [157]. Regarding clinical isolates of *P. aeruginosa* that exhibited resistance to carbapenem and aminoglycosides, a previously mentioned pharmacokinetic (PK) model evaluated the optimal dosage of tobramycin and imipenem [156]. The administration of aminoglycosides at significantly elevated dosages, in conjunction with continuous renal clearance techniques, was employed as a therapeutic approach to address infections caused by XDR *P. aeruginosa*, with the aim of mitigating the risk of renal toxicity. The results exhibited noteworthy rates of survival, despite the limited sample size of [160,161] individuals. In instances of severe or profound infections induced by MDR/XDR *P. aeruginosa*, alternative approaches for delivering aminoglycosides may be utilized. This is particularly applicable to conditions like pneumonia or meningitis. The administration of amikacin through inhalation for the treatment of pneumonia enables the attainment of elevated drug concentrations specifically at the infection site, such as the epithelial lining fluid (ELF). This approach also prevents excessive systemic exposure to the drug, which could potentially lead to systemic toxicity. Nevertheless, the utilization of inhaled antibiotics (specifically polymyxins or aminoglycosides) is recommended solely as an adjunctive therapy for infections caused by Gram-negative bacilli that exhibit susceptibility exclusively to aminoglycosides or polymyxins when used in conjunction with other drugs administered systemically [162]. Meningitis is a challenging infection to manage. The effectiveness of intravenous aminoglycosides is constrained by their limited ability to penetrate the central nervous system, resulting in insufficient and suboptimal concentrations at the infection site. In instances of this nature, the administration of intraventricular aminoglycosides may be necessary. A case of post-surgical meningitis caused by PDR *P. aeruginosa* was effectively managed through the administration of intravenous cefepime via continuous infusion, in combination with intravenous and intraventricular amikacin [163]. Despite the fact that the strain exhibited a minimum inhibitory concentration (MIC) of 32 mg/L for amikacin, the attainment of concentrations as high as 200 mg/L in the central nervous system proved to be effective in resolving the infection. Enzymatic modification of aminoglycosides through aminoglycoside-acetyltransferases (AAC), aminoglycoside-adenyltransferases (AAD), and aminoglycoside-phosphotransferases (APH) represents the prevailing resistance mechanism in *P. aeruginosa* for the aminoglycoside class of antibiotics. Furthermore, these enzymes can also be encoded on mobile genetic elements, thereby facilitating their widespread dissemination. A study was conducted to analyze 137 *P. aeruginosa* isolates collected from the University Hospital in Cumana, Venezuela. The most commonly observed genes in these isolates were *aphA1*, *aadB*, and *aac* (6′)-Ib. Among these genes, the last enzyme, *aac* (6′)-*Ib*, was found to catalyze N-acetylation at the 6′ position, which is a common mechanism employed by *P. aeruginosa* to modify aminoglycosides. The mean resistance observed for gentamicin was 32.6%, amikacin exhibited a resistance of 24.6%, and tobramycin demonstrated a resistance of 29.9% [164]. Previous research conducted in Latin American nations has revealed a prevalence of resistant strains ranging from 27.8% to 42.0% for gentamicin and 16.3% to 28.9% for amikacin [165].

### 5.5. Fosfomycin

The reemergence of intravenous fosfomycin as a treatment for infections caused by MDR bacteria is attributed to its remarkable bactericidal activity against various species, including MDR *P. aeruginosa* [166,167], as demonstrated in laboratory studies. Several additional experiments have evaluated the utilization of fosfomycin in conjunction with β-lactams, aminoglycosides, or colistin. In addition, this purportedly “ancient” antibiotic has been linked to more contemporary antibiotics such as ceftazidime–avibactam and ceftolozane–tazobactam [155,168]. The patient with XDR *P. aeruginosa* meningitis [169] was effectively treated by administering a combination of Ceftolozane–tazobactam and fosfomycin at a dosage of 4 g every 6 h.

Several studies have been conducted to evaluate different dosage regimens of fosfomycin in combination with carbapenem for the treatment of non-MDR and MDR *P. aeruginosa* clinical isolates, based on the attainment of the pharmacokinetic/pharmacodynamic (PK/PD) target. One study utilized a Monte Carlo simulation to assess the probability of achieving the desired outcome for different doses and durations of carbapenem and fosfomycin administration [170]. In the case of non-MDR *P. aeruginosa* isolates, the administration of a carbapenem and fosfomycin through continuous infusion at a rate ranging from 16 to 24 g per day resulted in the most favorable pharmacokinetic/pharmacodynamic ratios. The clinical isolates examined in this study, which took place in Thailand, exhibited remarkably elevated fosfomycin minimum inhibitory concentration (MIC) values. Therefore, it is not possible to extrapolate these results to other settings [170]. Further clinical series and trials are necessary to determine the future role of fosfomycin in these infections, encompassing optimal dosage and potential combinations. The effectiveness of fosfomycin IV against recurrent *P. aeruginosa* infections was studied in a phase I clinical study. Fosfomycin IV interferes with the formation of cell walls by inhibiting peptidoglycan assembly. A total of 30 healthy participants between the ages of 18 and 45 were recruited for the trial, and they were randomized to one of three treatment sequences, each lasting between 18 and 26 days (NCT02178254).

### 5.6. Newly Discovered Antimicrobial Agents

When present, murepavadin prevents Gram-negative bacteria from transporting LPS, as shown in a clinical trial (NCT02096315), and represents new class of antibiotics known as outer membrane protein targeted antibiotics (ompTAs) [171]. Murepavadin attaches to the outer membrane protein lipopolysaccharide transport protein D (LptD), which is important in the production of lipopolysaccharides in Gram-negative bacteria. By inhibiting LptD’s ability to transport LPS, it alters lipopolysaccharides and ultimately leads to cell death [171]. Murepavadin’s potent bactericidal properties were shown in vitro with 1219 *P. aeruginosa* isolates, many of which were multidrug resistant, collected from 112 hospitals in the US, China, and Europe. These tests determined that murepavadin’s MIC50 against numerous isolates of *P. aeruginosa* was 0.12 mg/L [172].

## 6. Therapeutic Strategies for *P. aeruginosa* Treatment

The process of developing novel antibiotics is highly time-consuming and constrained, thereby impeding the progress of therapeutic strategies for challenging *P. aeruginosa* infections. These therapeutic approaches can function independently or in conjunction to counteract infections. They encompass the suppression of quorum sensing and bacterial lectins, phage therapy, vaccine strategies, and antimicrobial photodynamic therapy [171,173]. Table 2 displays the treatment options for varying degrees of drug resistance in *P. aeruginosa*.

Despite the perplexing resistance mechanisms exhibited by *P. aeruginosa*, this pathogen remains a significant threat in clinical settings. However, there is optimism that through the utilization of emerging knowledge and its practical implementation, we can effectively combat this pathogen. Several emerging therapeutic approaches currently being investigated include combinatory therapy, the utilization of antimicrobial peptides (AMPs), bacteriophage therapy, and nanoparticles. Table 3 displays the therapeutic approaches for multidrug resistant MDR and extensively drug-resistant XDR *P. aeruginosa* infections, along with various mechanisms of action utilized.

### 6.1. Combinatory Therapy

The utilization of various drug combinations offers a promising approach to address the issue of antimicrobial resistance. This strategy involves employing a multi-targeted attack, taking advantage of their synergistic effects, and reducing the required dosage and associated side effects. In the case of *P. aeruginosa*, which exhibits multiple mechanisms of antibiotic resistance, combinatory therapy has the potential to be more effective, as indicated in Table 3 and Figure 3.

Ceftolozane–tazobactam (5th generation cephalosporin combined with a β-lactamase inhibitor)

The efficacy of this combination has been demonstrated in effectively combating *P. aeruginosa*, including MDR/XDR strains, by inhibiting the activity of penicillin-binding proteins (PBPs) [180]. The unique three-dimensional structure of the compound makes it more resistant to hydrolysis by AmpC β-lactamase in comparison to commonly employed cephalosporins, resulting in increased stability. Despite the emergence of resistance mechanisms involving class D β-lactamases and oxacillinases (OXA), as well as reduced membrane permeability due to OprD porin mutation and increased expression of efflux pumps [101,181], Ceftolozane–tazobactam remains a viable treatment option for multidrug-resistant/*P. aeruginosa* infections. Additionally, it can be considered as the preferred initial drug for carbapenem-resistant *P. aeruginosa* (CRPA) infections [7]. The combination of treatments has demonstrated positive results in severe infections occurring in intensive care units [182], as well as infections caused by *P. aeruginosa* in individuals with hematologic malignancies [183]. It has also shown effectiveness in treating complicated urinary tract infections and intra-abdominal infections [184], particularly when there is a high prevalence of *P. aeruginosa* in the microbiological intent-to-treat population. For uncomplicated urinary tract infections (UTIs) caused by drug-resistant *P. aeruginosa* DTRPA strain, it is recommended to administer a high and immediate dose therapy of this combination [185]. Severe infections induced by CRPA have demonstrated significant advantages when treated with CT (combinatory therapy) in comparison to the utilization of polymyxin or other combinations based on aminoglycosides. The European Society of Clinical Microbiology and Infectious Diseases (ESCMID) guidelines recommend the use of CT if the strain exhibits susceptibility in vitro. Caution should be exercised when determining the ideal dosage, particularly in cases of renal impairment [186] and in situations involving high-inoculum infections where the development of resistance may occur [13].

Ceftazidime–avibactam

Avibactam exhibits potent activity against resistance mechanisms, including extended-spectrum beta-lactamases (ESBLs), AmpC cephalosporinases, and OXA enzymes, thereby enhancing the efficacy of this combination. This combination has undergone testing in complex urinary tract infections (UTIs) and hospital-acquired pneumonia (HAP), demonstrating a notable degree of safety and effectiveness in the treatment of infections caused by 4 to 35% of isolates classified as MDR/XDR, as well as nearly all drug-resistant *P. aeruginosa* DTRPA strains [187]. Following the administration of colistin, it was observed that this particular combination exhibited the highest efficacy against subsets of pathogens that had developed resistance in patients hospitalized with pneumonia in Western Europe during the year 2020 [8]. The resistance to this combination may be ascribed to alterations in porin or efflux pump activity, or it could be a result of acquired cross-resistance against the ceftolozane–tazobactam combination. While the efficacy of this drug combination appears to be promising, it is advisable to exercise caution when administering it in cases of severe infections.

Imipenem–Cilastin–Relebactam (carbapenem + dehydropeptidase inhibitor + non-β lactam bicyclic diazabicyclooctane β-lactamase inhibitor)

This combination exhibits efficacy in strains that possess resistance against porins, as its mechanism of action is not reliant on the frequently implicated multidrug efflux pumps (MexA-MexB-OprM) [188]. Based on the findings of the SMART European surveillance, it was observed that there was imipenem susceptibility detected in isolates that were initially deemed non-susceptible [189]. The strain of *P. aeruginosa*, which was obtained from respiratory tract infections in intensive care unit settings, intra-abdominal injuries, and urinary tract infections, exhibited a significantly higher susceptibility to the combination treatment compared to imipenem alone. Despite exhibiting lower activity compared to ceftolozane–tazobactam, the imipenem–cilastin–relebactam combination may prove beneficial in the treatment of infections that have shown resistance to the ceftolozane–tazobactam combination [190]. Given that the majority of data regarding the effectiveness of this drug combination are derived from in vitro studies, its definitive role in clinical practice remains to be fully established.

Meropenem–Vaborbactam (Carbapenem + cyclic boronic acid β-lactamase inhibitor)

The distinguishing characteristic of this combination resides in its strong attraction to serine residues. Consequently, it forms covalent bonds with the β-lactamase enzyme, thereby competitively inhibiting its hydrolytic activity. The combination has been observed to enhance the sensitivity to meropenem in strains that exhibit lower sensitivity to meropenem alone. This finding highlights the potential significance of the combination in treating *P. aeruginosa* infections [191]. In certain instances, characterized by resistance mediated by KPC (Klebsiella pneumonia carbapenemase), this particular combination has demonstrated its efficacy against MDR/XDR strains [192]. However, its activity against MBL or OXA is reduced, and the combination of meropenem and vaborbactam does not exhibit a wider range of effectiveness compared to meropenem alone. This is because the resistance mechanisms of MBL or OXA are not overcome by the addition of vaborbactam.

Cefepime–Taniborbactam (4th generation cephalosporin + Boronic acid β-lactamase inhibitor)

*P. aeruginosa* has undergone evolutionary changes that have led to the acquisition of a notable resistance to cefepime through the upregulation of efflux pumps and increased synthesis of chromosomal AmpC enzymes. To enhance the effectiveness of cefepime, researchers have introduced a new compound called taniborbactam. This combination is currently being tested in phase 1 and phase 2 trials to determine if it is at least as effective as existing treatments. The trials involve 211 participants. Interestingly, taniborbactam is the first beta-lactamase inhibitor (BLI) that can competitively inhibit all metallo-beta-lactamases (MBLs), except for the IMP-type. It achieves this by directly inhibiting Ambler class A, B, C, and D enzymes. Therefore, it can be regarded as a potential combination in the production of CRPA and MBL isolates [193].

Cefepime–Zidebactam (4th generation cephalosporin + β-lactam enhancer antibiotic)

The observed activity of this combination is achieved by inhibiting the activity of penicillin-binding protein 3 (PBP3) through cefepime, and penicillin-binding protein 2 (PBP2) through zidebactam. The compound has demonstrated significant efficacy against *P. aeruginosa* isolates, including those that exhibit high levels of efflux pump production. This phenomenon can be attributed to the unaffected activity of the organism in question, regardless of the expression of ESBLs (Extended-Spectrum Beta-Lactamases), OXA-48-like carbapenemases, and MBL (Metallo-Beta-Lactamase) carbapenemases. While in vitro studies have shown resistance against this combination, it remains a promising choice for isolates where local resistance mechanisms are of greater concern [194].

Meropenem–Nacubactam (Carbapenem + diazabicyclooctane (DBO) type β- lactamase inhibitor)

While this combination exhibits the inhibition of penicillin-binding protein 2 (PBP2) and demonstrates activity against AmpC hyperproducing and KPC-producing *P. aeruginosa*, its efficacy is more pronounced against Enterobacterales [195]. The following Table 4 presents the effectiveness of different drug combinations, considering their performance in in vitro time-killing (TK) or pharmacokinetic/pharmacodynamic (PK/PD) studies. The combinatorial ratios, as determined by the minimum inhibitory concentration (MIC) of individual strains, were assessed using in vitro models. The level of synergy was classified as low, moderate, or high. It is crucial to take into account the diminished impact of both combination therapies and monotherapies as a result of the continuous release of planktonic cells from biofilms in in vivo situations. Consequently, the effectiveness of any drug combination therapy is significantly influenced by the maturity of the biofilm, necessitating clinical trials to optimize therapeutic outcomes.

### 6.2. Antimicrobial Peptides

Antimicrobial peptides (AMPs) are bioactive substances that are remarkably biocompatible and less likely to cause bacterial resistance to evolve [211,212]. AMPs encompass a wide range of naturally derived or artificially synthesized small peptides that exert their effects on relatively non-specific targets, primarily within cells. These peptides exhibit a reduced propensity to induce the development of resistance [213]. This emerging therapeutic modality exhibits numerous advantages, including a high bactericidal potency at micromolar concentrations [214], a mechanism of action that targets multiple sites, and a rapid onset of effect [215]. Propelled by the emergence of antimicrobial peptides (AMPs), a novel category of peptides exhibiting remarkable target specificity and heightened responsiveness, known as “selectively targeted AMPs” (STAMPs), have been devised and are currently garnering escalating significance [216]. Antimicrobial peptides (AMPs), including human cathelicidin peptide (LL-37), colistin (derived from Bacillus polymyxa var colistinus), and colistin-derived AMPs (AA139 and SET-M33), have been suggested as potential therapeutic agents for combating drug-resistant *P. aeruginosa* infections. These substances exhibit bactericidal properties without promoting the development of resistance, and they also demonstrate activity against biofilms [217].

### 6.3. Phage Therapy

Bacteriophage therapy is a recently developed and highly innovative therapeutic approach aimed at addressing the issue of antibiotic resistance. Among the two classifications of bacteriophages, namely lytic and temperate, it is only the lytic phage that can be utilized in clinical scenarios. The substance exhibits specific binding affinity towards the external membrane of the bacterial cells, facilitating the transfer of its genetic material into the recipient cell. The phage replicates within the host cell by utilizing host proteins, and the resulting offspring phages migrate and exhibit bactericidal activity towards specific targets, such as flagella and pili organs of the bacteria. These substances can be administered via systemic injection or applied topically to wounds to exert their bactericidal properties. Various combinations of phages, referred to as phage cocktails, are currently undergoing testing. These experiments have shown enhanced effectiveness in treating *P. aeruginosa* strains that are resistant to conventional treatments (as indicated in Table 5). Experiments are being conducted to evaluate the efficacy of bacteriophages that target the virulent factors of *P. aeruginosa* strains. *P. aeruginosa* releases a variety of extracellular toxins that have been documented to induce significant harm by disrupting blood clotting and causing tissue death. Bacteriophages capable of modifying these exotoxins and inducing their inactivation are currently under development [218,219,220]. Bacteriophages have the potential to eliminate *P. aeruginosa* biofilms through mechanisms such as extracellular matrix degradation, enhanced permeability of the inner biofilm layer, and suppression of quorum-sensing activity, as documented in a study [221]. Pili and fimbriae serve as the primary factors for bacterial adherence. Bacteriophages are being developed to inhibit the expression of these factors [222]. Mono-phage, phages-cocktail, phage-derived enzyme (lysin), bio-engineered phage, and phage in combination with antibiotics are all forms of phage therapy [223]. A thorough search of the literature revealed several case study reports and compassionate uses for severely ill patients at specialized facilities, but despite encouraging outcomes, recent clinical trials based on evidence will be required. Due to PP1131’s lack of efficacy vs. SOC, the first randomized controlled trial using a combination of natural lytic *P. aeruginosa* phages for the topical treatment of infected burn wound patients was discontinued in January 2017 [224]. Phage therapy poses significant challenges due to safety considerations, including the presence of phage-neutralizing antibodies and immune responses, as well as the potential for resistance development via the CRISPR-Cas system.

## 7. Conclusions

MDR/XDR Infections caused by *P. aeruginosa* pose a significant challenge within healthcare environments. The antibiotic resistance observed in *P. aeruginosa* is attributed to mechanisms that are either innate, acquired, or adaptive in nature. *P. aeruginosa* is a remarkably adaptable pathogenic microorganism possessing an intricate regulatory network, which is among the most intricate in the bacterial kingdom. The presence of interference can potentially lead to secondary effects on the cellular physiology and result in complications for infected individuals. A more comprehensive understanding of the host bacteria as a cohesive system in its response to treatment is imperative in order to formulate innovative therapeutic approaches to combat this exceptionally formidable human pathogen. Multiple antibiotic resistance mechanisms play a role in the emergence of strains, rendering traditional antibiotics ineffective in the treatment of MDR/XDR *P. aeruginosa* infections. Moreover, the process of biofilm formation in *P. aeruginosa* persister cells is accountable for the persistence and resistance of infections in individuals with cystic fibrosis.

Nevertheless, clinical *P. aeruginosa* exhibits a remarkable ability to evolve novel resistance mechanisms against both established and emerging antibiotics. This poses significant challenges and potential threats to public health. The non-antibiotic therapeutic strategies, specifically phage therapy and Antimicrobial peptides, have demonstrated noteworthy antimicrobial properties against antibiotic-resistant strains of *P. aeruginosa* in laboratory settings or animal experiments. Future investigations should prioritize the exploration of innovative methodologies aimed at mitigating adverse impacts and enhancing the levels of safety and effectiveness observed in clinical trials. *P. aeruginosa* utilizes a diverse antibiotic resistance strategy. The optimal approach for future treatments is expected to involve combinational therapies targeting this highly virulent pathogen. This strategy entails the integration of novel treatments with conventional antibiotics to achieve the successful eradication of the pathogen in immunocompromised patients who are particularly susceptible to its detrimental effects.

## Figures and Tables

**Figure 1 pharmaceuticals-16-01230-f001:**
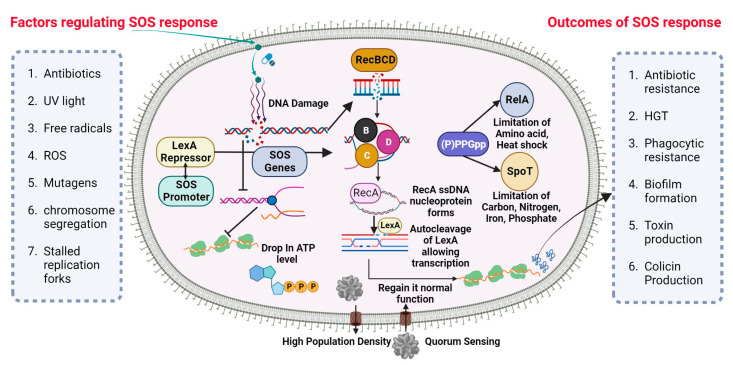
SOS DNA repair mechanism in *P. aeruginosa*. Nucleic acid inhibitor antibiotic nitration can stimulate the SOS response in *P. aeruginosa* via the synthesis of the RecA gene from RecBCD subunit and upregulating Lex-containing TisAB, leading to an ATP level drop and the downregulation of important cellular functions. RecA filaments merge and trigger the SOS response. When LexA and RecA-ssDNA nucleoprotein filament connect, LexA’s latent protease activity is activated, leading to LexA’s autocleavage. After LexA is autocleaved and rendered inactive, the SOS gene’s transcription is triggered, inducing a global transcriptional response.

**Figure 2 pharmaceuticals-16-01230-f002:**
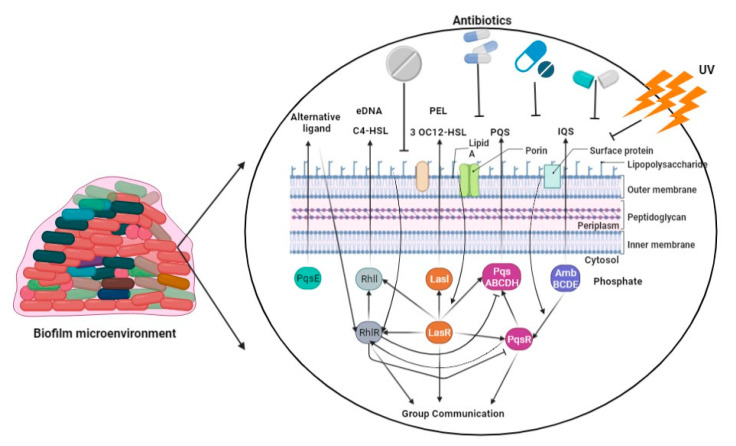
The *P. aeruginosa* employs four interwoven quorum-sensing loops for biofilm formation using LasI and LasR, RhlI, PqsE and RhlR, PqsABCDH and PqsR, and AmbBCDE.

**Figure 3 pharmaceuticals-16-01230-f003:**
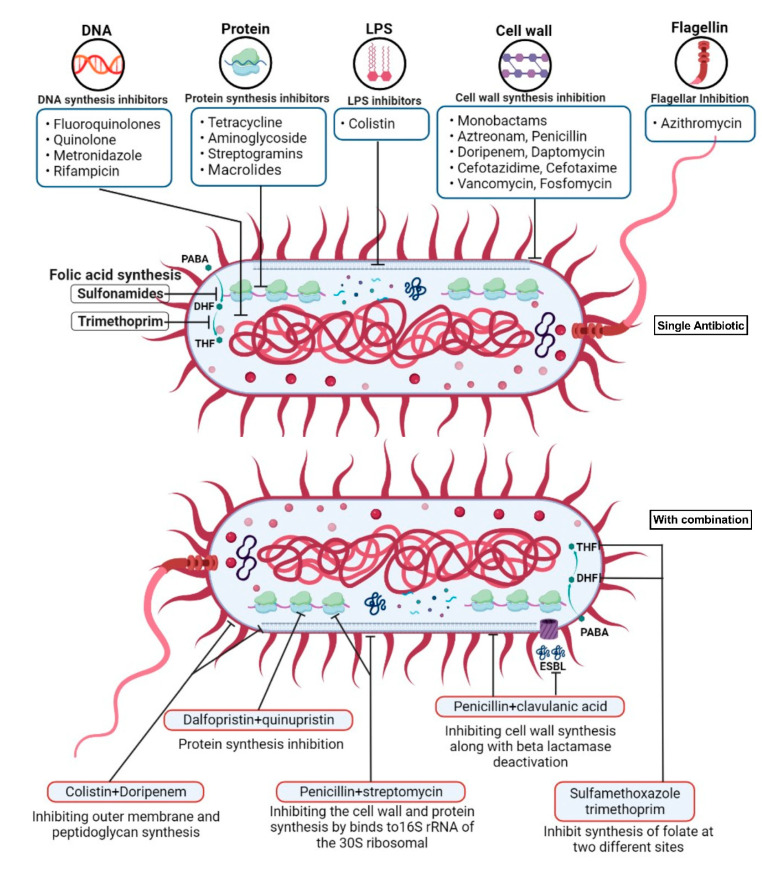
Diagrammatic representation showing mechanisms of action of single antibiotics and antibiotic with combinations.

**Table 1 pharmaceuticals-16-01230-t001:** Key genes that are known to be associated in *P. aeruginosa* mutational antibiotic resistance [2,11].

Responsible Genes	Mechanism of Resistance	Associated Antibiotics
*gyrA*	Target modification of Quinolones (DNA gyrase)	Fluoroquinolones
*gyrB*	Target modification of Quinolones (DNA gyrase)	Fluoroquinolones
*parC*	Target modification of Quinolones (DNA topoisomerase IV)	Fluoroquinolones
*parE*	Target modification of Quinolones (DNA topoisomerase IV)	Fluoroquinolones
*phoQ*, *cprS*, *colR*, *colS*, *pmrA*, *pmrB*	Modification of Lipopolysaccharide (addition, 4-amino-4-deoxy-L-arabinose moiety to the lipid A portion)	Polymyxins
*parR*	Modification of Lipopolysaccharide (addition, 4-amino-4-deoxy-L-arabinose moiety to the lipid A portion)	Polymyxins
	Hyperproduction of efflux-mediated genes (MexEF-OprN)	Fluoroquinolones
	Downregulation of OprD	Imipenem, meropenem
	Hyperproduction of efflux-mediated genes (MexXY)	Aminoglycosides, cefepime
*parS*	Modification of Lipopolysaccharide (addition, 4-amino-4-deoxy-L-arabinose moiety to the lipid A portion)	Polymyxins
	Downregulation of OprD	Imipenem, meropenem
	Hyperproduction of efflux-mediated genes (MexEF-OprN)	Fluoroquinolones
	Hyperproduction of efflux-mediated genes (MexXY)	Fluoroquinolones
*mexR*	Hyperproduction of efflux-mediated genes (MexAB-OprM)	Fluoroquinolones
*nfxB*	Hyperproduction of efflux-mediated genes (MexCD-OprJ)	Fluoroquinolones, cefepime
*mexS*	Hyperproduction of efflux-mediated genes (MexEF-OprN)	Fluoroquinolones
	Downregulation of OprD	Imipenem, meropenem
*mexT*	Hyperproduction of efflux-mediated genes (MexEF-OprN)	Fluoroquinolones
*cmrA*, *mvaT*, *PA3271*	Hyperproduction of efflux-mediated genes (MexEF-OprN)	Fluoroquinolones
*mexZ*, *PA5471.1*, *amgS*	Hyperproduction of efflux-mediated genes (MexXY)	MexXY hyperproduction
*oprD*	Inactivation of Porin channels	Imipenem, meropenem
*ampD*, *ampDh2*, *ampDh3*, *ampR*, *dacB*, *mpl*	Hyperproduction of AmpC	Ceftazidime, cefepime, piperacillin–tazobactam
*ftsI*	Target modification (PBP3)	Ceftazidime, cefepime, piperacillin–tazobactam
*fusA1*	Target modification of Aminoglycoside (elongation factor G)	Aminoglycosides
*glpT*	Transporter protein inactivation GlpT	Fosfomycin
*rpoB*	Rifampin target modification, RNA polymerase β-chain	Rifampin

**Table 2 pharmaceuticals-16-01230-t002:** Treatment options of different levels of drug resistant *P. aeruginosa*.

Treatment Options	MDRPA (Multidrug Resistant *P. aeruginosa*)	DTRPA (Difficult-to-Treat Resistant *P. aeruginosa*)	XDRPA (Extensively Drug Resistant *P. aeruginosa*)	CRPA (Carbapenem Resistant *P. aeruginosa*)	MBL-Producing-CRPA (Metallo-β-lactamase)	*P. aeruginosa* Resistant to Ceftolozane–tazobactam Combination
COMBINATORY THERAPY						
Ceftolozane–tazobactam	√	√	√ (High, immediate dose)	√ (1st line drug)		
Ceftazidime–avibactam	√	√	√			
Imipenem–Cilastin–Relebactam	√	√				√
Meropenem–Vaborbactam	√		√			
Cefepime–Zidebactam	√				√	
Cefepime–Taniborbactam				√	√	
Fosfomycin (along with other combinations)	√	√				
NEWER DRUGS						
Cefiderocol	√	√		√	√	√
ANTIMICROBIAL PEPTIDES (AMPs)						
AMPs–colistin	√					

**Table 3 pharmaceuticals-16-01230-t003:** Summary of the strategies discovered in recent years for the treatment of MDR/XDR *P. aeruginosa* infections and different mechanisms of action.

Therapeutic Strategies	Mechanisms	Advantages	Reference
Antibiotic combinations	Combinations with antibiotic to antibiotic or other substances to destroy biofilms and prevent the antibiotic resistance.	combinations provide an excellent solution to the resistance to antimicrobials as it explores the usage of multi-targeted attack, synergism, and reduced dosage and side effects	[174,175]
Antimicrobial peptides (AMPs)	Induce membrane disruption, leading to cell lysis and death. AMPs enter into the cells without membrane disruption and inhibiting intracellular function by binding to nucleic acid.	Inhibits the quorum-sensing system rapid killing kinetics, low levels of induced resistance, low toxicity to host	[176,177]
Phage therapy	Viral assembly, destroy extracellular matrix by encode enzymes.	Highly specific, replication at infection site without effects commensal flora. Easy administration, easy delivery.	[178,179]

**Table 4 pharmaceuticals-16-01230-t004:** Summaries of antibiotic combination therapeutic strategies for treatment of *P. aeruginosa*.

SR NO.	Drug Combination		Numder of PA Strains	Synergy	Source
1.	Carbapenems + Aminoglycosides	Imipenem + Amikacin	27 87	Low (TK) High (PK/PD)	[196,197]
		Imipenem + Tobramycin	3	Moderate (TK)	[198]
		Meropenem + Amikacin	63 3	Moderate (TK)	[130,199]
2.	Carbapenems + Fluoroquinolones	Imipenem + Levofloxacin	5	Moderate	[200]
		Meropenem + Ciprofloxacin	17	Low	[201]
3.	Fluoroquinolones + Polymyxins	Ciprofloxacin + Colistin	2 17	No synergy (PD) High	[145,201]
4.	Polymyxins + Carbapenems	Colistin + Doripenem	3	Moderate	[202,203]
		Colistin + Imipenem	2	Moderate (TK)	[202]
		Colistin + Meropenem	7 17	Moderate Low	[145,204]
5.	Cephalosporin + Aminoglycoside	Ceftazidime/avibactam + Amikacin	3 21	Low Moderate (TK)	[205]
		Ceftolozane/tazobactam + Amikacin	4 20	No synergy Moderate (TK)	[206,207]
6.	Cephalosporin + Polymyxin	Ceftolozane/tazobactam + Colistin	4	Moderate	[201,208]
		Ceftazidime + Colistin	2	Moderate (PD)	
7.	Cephalosporin + Monobactam	Ceftolozane/Tazobactam + Aztreonam	4	No synergy	[209]
8.	Polymyxin B + Tetracycline	Polymyxin B + Doxcycline	3	Moderate synergy (TK)	[210]

**Table 5 pharmaceuticals-16-01230-t005:** Clinical trials in humans of phage therapy to treat resistant *P. aeruginosa* strains.

Trial Number	Phase/Participants	Study Design	Type of Infection	Test Therapy	Result
NCT03140085	P2,3/113	Cohort (Randomized, Double-blinded, Parallel 3 arm)	Urinary tract infections in patients undergoing trans-urethral resection of the prostate.	Intravesical bacteriophage treatment with PYO phage.	Intravesical bacteriophage therapy was non-inferior to standard-of-care antibiotic treatment, but was not superior to placebo bladder irrigation in treating UTIs in patients undergoing TURP. Bacteriophage safety profile seems to be favorable.
NCT04803708	P I/IIa/20	Cohort (Randomized, Double-blinded)	Non-infected and infected diabetic foot ulcers with *P. aeruginosa*, Staphylococcus aureus and/or Acinetobacter baumanni.	Topical bacteriophage cocktail (TP-102).	(Not published)
NCT01818206	-/60	Interventional	Cystic fibrosis with *P. aeruginosa* infection.	A cocktail of 10 bacteriophages.	(Not published)

## Data Availability

Data sharing is not applicable.

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
