# Peer review of "Age of Antibiotic Resistance in MDR/XDR Clinical Pathogen of Pseudomonas aeruginosa"

_pharmaceuticals, 2023, doi:10.3390/ph16091230_

Round 1

Reviewer 1 Report

The authors present an interesting review paper on age of antibiotic resistance in MDR/XDR clinical pathogen of Pseudomonas aeruginosa.

But some minor changes or clarifications should be solved by authors, regarding the present manuscript form in order to be published in pharmaceuticals. Some recommendations can be found as follows:

-          The english must be checked by an expert.

-          Title : remove the word « garimas »

-          In the Title and throughout in the manuscript, the name of bacteria should be written in the italics forms.

-          Table 1 ad the reference for each mechanisms

   The english must be checked by an expert.

Author Response

The authors present an interesting review paper on age of antibiotic resistance in MDR/XDR clinical pathogen of Pseudomonas aeruginosa.

But some minor changes or clarifications should be solved by authors, regarding the present manuscript form in order to be published in pharmaceuticals. Some recommendations can be found as follows:

*** We made changes based on your suggestions.

Point 1. The english must be checked by an expert.

*** The English language was checked by an expert.  

Point 2. Title: remove the word « garimas »

*** We removed ‘garimas’ from the title. It was typo.

Point 3. In the Title and throughout in the manuscript, the name of bacteria should be written in the italics forms.

*** We have change the bacterial and gene names in italics.

Point 4. Table 1 ad the reference for each mechanisms

*** We have added the references for the table 1.

Reviewer 2 Report

The review: Age of antibiotic resistance in MDR/XDR clinical pathogen of  Pseudomonas garimas aeruginosa by Ashish Kothari and co-authors addresses the clinically significant problem of multidrug-resistant and extensively resistant strains of Pseudomonas aeruginosa. The authors characterize the various resistance mechanisms developed by P. aeruginosa, the epidemiology of MDR strains and their clinical impact, the antibiotics used in MDR P. aeruginosa infections, and the bactericidal effectiveness of antibiotics used to treat infections caused by P. aeruginosa. They point to new therapeutic strategies to combat P. aeruginosa infection.

The structure/layout of the manuscript largely mirrors that of an earlier publication by Spanish researchers [Horcajada JP, Montero M, Oliver A, Sorlí L, Luque S, Gómez-Zorrilla S, et al. Epidemiology and Treatment of Multidrug-Resistant and Extensively Drug-Resistant Pseudomonas aeruginosa Infections. Clin Microbiol Rev. 2019 Sep 18;32(4)], quoted in the text (item 11). The authors also use the same subsection titles. Is it so difficult to use your formulation of the names of subsections?  However, the manuscript's content is extended and more extensive with issues related to the potential impact of the SOS response in bacteria on the increase of their drug resistance, aspects of new drugs, antibacterial peptides, and phage therapy.

My substantive comment concerns subsection 2.4. Antibiotic resistance by SOS response - the authors try to describe the complex SOS response mechanism in bacteria very synthetically, and this generates shortcuts that create errors.
Namely:

- Polymerases IV and V are not synthesized at the site of DNA damage but act at the site of DNA damage

- Translesion Synthesis includes the activity of specialized TLSs of three polymerases (pol II, pol IV, and polV). DNA polymerase II prevents "mutational catastrophe."

- In the description of the process, the role of RecA and LexA proteins should be given in one sentence, making it easier to understand Figure.1.

- Within Figure .1. - typo in "Drop ATP level."

- In the description under Fig.1. - ...a worldwide transcriptional ...

I suggest replacing: global, general, or widespread

- T/A systems - if we mention it, the abbreviation should be explained

  - in turn, in the text, listing the classes of polymerases Y or C is not needed at this point, and if we do, we write the letter names of the enzymes without quotation marks because they result from the systematic classification.

General remarks apply to the entire text:

- sometimes words with capital letters appear in the middle of the sentence - lines: 232, 340

- quotation marks are often overused - lines: 304, 306, 310

- abbreviations of gene names should be written in italics - lines 150, 152, 154

- lines: 368-371 - sentence correction

Author Response

My substantive comment concerns subsection 2.4. Antibiotic resistance by SOS response - the authors try to describe the complex SOS response mechanism in bacteria very synthetically, and this generates shortcuts that create errors.

*** We made changes based on your suggestions.

Point 1. Polymerases IV and V are not synthesized at the site of DNA damage but act at the site of DNA damage

*** Thank you so much for pointing such important details in the manuscript, I have removed the unwanted mistakes according to suggestions.

Point 2. Translesion Synthesis includes the activity of specialized TLSs of three polymerases (pol II, pol IV, and polV). DNA polymerase II prevents "mutational catastrophe."

*** We have added three polymerases (pol II, pol IV, and polV).

Point 3. In the description of the process, the role of RecA and LexA proteins should be given in one sentence, making it easier to understand Figure.1.

*** Corrected!  

Point 4. Within Figure .1. - typo in "Drop ATP level."

*** Corrected!  

Point 5. In the description under Fig.1. - ...a worldwide transcriptional ...I suggest replacing: global, general, or widespread

*** Removed the word “worldwide” and changed with global. 

Point 6. T/A systems - if we mention it, the abbreviation should be explained

*** Abbreviated and specified!   

Point 7. in turn, in the text, listing the classes of polymerases Y or C is not needed at this point, and if we do, we write the letter names of the enzymes without quotation marks because they result from the systematic classification.

*** Edited accordingly!   

General remarks apply to the entire text:

Point 8. sometimes words with capital letters appear in the middle of the sentence - lines: 232, 340

*** Corrected!  

Point 9. quotation marks are often overused - lines: 304, 306, 310

*** Corrected!  

Point 10. abbreviations of gene names should be written in italics - lines 150, 152, 154

*** Corrected!  

Point 11. lines: 368-371 - sentence correction

*** Corrected!  

Reviewer 3 Report

The manuscript "Age of antibiotic resistance in MDR/XDR clinical pathogen of Pseudomonas garimas aeruginosa" mainly discussed the antibiotic resistant mechanism of MDR/XDR P. aeruginosa and and available treatments for their infections.

Major suggestions:

1. There have been many reviews on the resistance mechanisms and therapeutic strategies. Can you explain why this review is new or telling new things?

2. The content of the article is broad but not in-depth. For the molecular mechanism of drug resistance in P. aeruginosa, the review only involves which mutations lead to   antibiotic resistance, or which enzymes and genes are involved in drug resistance, but does not clarify the specific mechanism. And for antibiotic agents for of MDR P. aeruginosa, this review only lists the drugs that are commonly reported, and does not list newly developed antibiotics or antibacterial drugs. Please provide some newly developed antibiotic agents and therapeutic strategies for P. aeruginosa.

3. The third part is titled MDR P. aeruginosa epidemiology, but most of the content is about the criteria for MDR, XDR, and PDR P. aeruginosa. It is suggested to add the content of MDR P. aeruginosa epidemiology.

4. Reference 6 was published in 2015. Please provide some more recent data on the combination medication resistance of P. aeruginosa MDR and XDR strains.

Minor editing of English language required

Author Response

The manuscript "Age of antibiotic resistance in MDR/XDR clinical pathogen of Pseudomonas aeruginosa" mainly discussed the antibiotic resistant mechanism of MDR/XDR P. aeruginosa and available treatments for their infections.

Major suggestions:

Point 1. There have been many reviews on the resistance mechanisms and therapeutic strategies. Can you explain why this review is new or telling new things?

*** In this current review article we have highlighted the AMR mechanism considering the most common pathogen P. aeruginosa, its clinical impact, epidemiology, and SOS-mediated resistance. We have further discussed the current therapeutic options against MDR/XDR P. aeruginosa infections those treatment options in clinical practice. Finally, other therapeutic strategies, such as bacteriophage-based therapy and antimicrobial peptides, have been described with clinical relevance. 

Point 2. The content of the article is broad but not in-depth. For the molecular mechanism of drug resistance in P. aeruginosa, the review only involves which mutations lead to   antibiotic resistance, or which enzymes and genes are involved in drug resistance, but does not clarify the specific mechanism. And for antibiotic agents for of MDR P. aeruginosa, this review only lists the drugs that are commonly reported, and does not list newly developed antibiotics or antibacterial drugs. Please provide some newly developed antibiotic agents and therapeutic strategies for P. aeruginosa.

*** Now we have provided some newly developed antibiotic information.

Point 3. The third part is titled MDR P. aeruginosa epidemiology, but most of the content is about the criteria for MDR, XDR, and PDR P. aeruginosa. It is suggested to add the content of MDR P. aeruginosa epidemiology.

*** We have added XDR in the heading “MDR/XDR P. aeruginosa epidemiology”

Point 4. Reference 6 was published in 2015. Please provide some more recent data on the combination medication resistance of P. aeruginosa MDR and XDR strains.

*** Resent data and citations were been added!

Round 2

Reviewer 3 Report

The author has conducted the corresponding modification and completed very well.